# Design and Validation of a New Tennis-Specific Reactive Agility Test—A Pilot Study

**DOI:** 10.3390/ijerph191610039

**Published:** 2022-08-15

**Authors:** Goran Munivrana, Igor Jelaska, Mario Tomljanović

**Affiliations:** 1Faculty of Kinesiology, University of Split, 21000 Split, Croatia; 2International Table Tennis Federation (ITTF), 1007 Lausanne, Switzerland

**Keywords:** racket sports, field-based test development, reliability and validity, practical feasibility

## Abstract

Agility is an important ability for tennis players, but there is an evident lack of studies focusing on the applicability of tennis-specific agility tests that capture a combination of the physical and cognitive agility performance. Therefore, the main aim of this study was to design and test measurement properties of the tennis-specific reactive agility test that would be feasible and practical for regular implementation. A total of 32 youth tennis players (21 males and 11 females; 10.85 ± 1.50 years) participated in this study. The measurement characteristics of the newly designed reactive agility tennis-specific (TS-RAN) test have been established by comparing it with two generic agility tests and with two tennis-specific pre-planned agility tests. The overall reliability of the new TS-RAN test can only be considered “moderate to good”, as although the results of participants showed good internal consistency, the within-subject reliability of the test proved to be unsatisfactory, since the participants showed a lack of performance consistency. This is not unexpected considering the very young age of the participants who performed the test for the very first time. The new TS-RAN test was highly projected (0.91) on the same single latent dimension, with the variance predominately explained by the tennis-specific agility tests. The test’s greatest strength is its high feasibility, since the test does not require any special set-up nor technical equipment, and that makes it practical for regular implementation in a practical setting. Further research studies are needed in order to confirm the test’s potential to be widely accepted and used.

## 1. Introduction

During the tennis rallies, players need to perform rapid accelerations, decelerations and changes of directions (CODs), also referred to as agility performance [1]. An average occurrence of CODs in tennis game varies from 2–4 per point, depending on the quality of players and the playing surface the match is being played on [2,3].

These rapid changes in direction are not only performed in a linear and lateral direction but are multidirectional in nature. All these multidirectional movements are performed as reactions to information received, in form of upcoming balls from opponents’ side of the tennis court, and depend on their speed, direction, rotation and flight path [4].

Although the definition of agility mainly refers to the ability of changing direction rapidly [5], it is obvious that in tennis and all racket sports, this ability cannot be considered as a pure physical skill [6]. The process of responding to the opponent’s actions during rallies in tennis and other racket sports includes and combines not just physical skills, but also cognitive and technical skills [7].

Therefore, it is recommended to use an assessment that captures a combination of the physical and cognitive (“reaction to a stimulus”) aspects of agility, which are both equally needed during a tennis match [1,4].

In tennis, in response to a primary visual stimulus, e.g., an approaching ball hit by an opponent, players have to accelerate/decelerate and constantly change direction while performing rapid sport-specific movements.

With all the above mentioned in mind, it is logical to presume that in order to provide the most realistic and comprehensive profile of agility performance during a tennis match, an agility test should represent the real game situations as closely as possible (e.g., response to a stimulus, inclusion of tennis racquet, tennis-specific footwork). It is also a reasonable assumption that the assessment of agility under sport-specific conditions should always provide the most complete profile of agility performance [8,9].

According to the existing literature, a solid number of agility tests with various levels of specificity have been developed and used in tennis over the years, but majority of the used tests were primarily change of direction (COD) tests [9,10,11,12,13,14,15]. Those tests did not really consider the cognitive, “reaction to a stimulus” aspect of agility, which would be a much better representative of the kind of agility performance needed in a real tennis match rally situation. The test used in the study of Ulbricht et al. [16] included both physical and cognitive aspects of agility in the test, but the duration, number of CODs (just one) and some other parameters used in the test made it more a tennis-specific sprint speed test than a real reactive agility one. It was not until very recently that a genuine tennis-specific reactive agility test (TAT) for monitoring tennis players has been designed by a group of Dutch authors [17]. The test has showed promising results, with solid test–retest reliability and concurrent validity in relation to a popular generic Spider Drill agility test [14]. However, although the TAT test has been designed to be used in a practical setting by sport scientists and coaches, the test still requires some technical equipment (FITLIGHT TrainerTM system) and setting to be arranged (cones and lights positions, etc.) for it to be successfully conducted.

On the other hand, the main aim of this study was to design a tennis-specific reactive agility test, to be conducted on a tennis court, aiming for it to be even more feasible and practical for implementation during training sessions than any of the existing tennis-specific agility tests. This test does not require any special technical equipment nor does it take time to arrange a specific setting for the test, as it uses the already existing lines on the tennis court. At the same time, the test contains all the prerequisites necessary for the test to be considered a reactive agility sport-specific test. That includes response to a visual stimulus, inclusion of tennis equipment (tennis racket and ball) and players/examinees using tennis-specific footwork/movement on the court. Testing metric characteristics of the newly designed tennis-specific agility test and evaluation of its key measurement properties would pertain to specific objectives of the study.

## 2. Materials and Methods

### 2.1. Participants

The sample consisted of 32 youth tennis players (21 males and 11 females; 10.85 ± 1.50 years). All participants were between 10 and 12 years of age and had been practicing tennis for at least 3 years at the moment of testing.

Thus, all participants were relative beginners to the world of tennis, but at the level advanced enough to have mastered all the basic strokes and footwork techniques of the tennis game. Majority of them have already regularly participated in national level tournaments for their age categories (U10, U12) and some have been highly ranked at the national ranking list for their age category.

The test results have not shown any statistically significant differences between genders, most likely due to the young prepuberty/early puberty age of the participants. It is obvious that hormonal, gender-related inputs have still not reached their full swing within this sample of youth participants. Therefore, the gender issue does not have a significant influence on the test results.

The testing protocol was conducted in full compliance with the Declaration of Helsinki and informed parental consent was obtained for all the participants of the study.

### 2.2. Measures/Variables

In this study, the researchers used several test instruments to evaluate the newly designed Tennis-Specific Reactive Agility New test (TS-RAN). For the evaluation of its reliability, three trials were performed by each of the participants. The factorial validity of the newly designed reactive agility tennis-specific test (TS-RAN) has been established by comparing it with two validated generic agility tests and with two newly designed tennis-specific pre-planned agility tests.

The five tests constituting the test battery appear as follows:

**Multi-direction agility “ABCD” test (****MDA “ABCD”)** [18] is a generic agility test, very similar in duration and movement pattern to the other popular and standardly used agility tests (T-TEST) [19]. *Equipment:* Tape and 1 stopwatch.*Space:* The task is carried out in a closed or open space minimum dimensions of 15 × 7 m. On the floor, there are two parallel lines (10 and 5 m long) connected at their bases with the 5 m long vertical line. (Figure 1).*Task description:* Participant is in a starting position at point A and at the moment of receiving a start signal the participant covers the distance from A to B and back with side steps, then sprints forward to the C point and sprints backwards back to A, from where he/she uses sidesteps to the point B and from there makes a final sprint run to D. The task is completed when the participant passes point D. *Measurer:* One measurer, standing at point D. *Assessment/Evaluation:* The result is a time measured in tenths of a second from the start signal until the completion of the task. Three measured attempts, all are recorded on a score form.**Steps to the side lateral agility (STSLA)** test [20] measures lateral (left to right and vice versa) movement ability, which is one of the most common movement patterns used by tennis players in the game. The test assesses lateral speed, agility and body control. The test is basically identical in duration and movement pattern to the commonly used Edgren side-step test [21].*Equipment:* Adhesive tape and 1 stopwatch.*Space:* The task is carried out in a closed or open space with the minimum dimensions of 5 × 2 m. Two parallel lines 1.5 m long are positioned on the floor at a distance of 4 m.*Task description:* On a course which is 4 m in length (Figure 2), participants start on the far left, touching/crossing the left parallel line with their left foot. When the participants receive a starting signal, they side-step 4 m to the right until their right foot touches/crosses the outside right line mark, then side-step to the left until their left foot touches/crosses the left outside line left mark, and do it back and forth as rapidly as possible, till the participants successfully completes six 4 m distances. The task is finished when the participants touch/cross the left parallel (starting/finishing) line for the third time.*Measurer:* One measurer, stands in front of participants, counting 6 distances and measuring time with a stopwatch.*Assessment/Evaluation:* The result is a time measured in tenths of a second from the start signal until the completion of the task. Three attempts are carried out and all are recorded on a score form.**Tennis-specific steps to the side lateral agility (TS-STSLA)** is a test designed to assess the specific agility in tennis and uses tennis equipment for a stroke. The test is in nature very similar to the test “Steps to the side—a test of lateral agility [19]” with the difference that on one side instead of crossing the foot over the line, a tennis kick is performed.*Equipment:* Adhesive tape, stopwatch, ball and tennis racquet.*Space:* Test is conducted on a tennis court.*Task description:* The test is almost identical to the “steps to the side” test, but in this case, the main goal is to cross the distance from the center of the baseline to the side “single” line and hit a vertically dropped ball by one of the measurers at the intersection of these two lines. Participants hit a forehand, so in the case of playing with the left hand, the test takes place on the other/left side of the field. Participants begin the test with the left foot (if left-handed, the test begins with the right foot) at the center line that divides the baseline into two equal parts. With a sideways movement they reach the place where the ball is dropped, hit the forehand and return back to the center line which they must touch with their left foot or cross over the line with their foot. The participant must hit 3 forehands and cross the distance 6 times, and the time stops when the participant crosses the center line on the baseline (Figure 3).*Measurer:* Two measures, one dropping the balls and the other with a stopwatch.*Assessment/Evaluation*: The result is the time measured in tenths of a second from the start signal to the end of the task. Three attempts are carried out and all are recorded on a score form.**Tennis-specific multi-directional agility (TS-MDA)** is a test designed to assess specific agility in tennis with a pattern of movement that simulates the actual situation in the game, with participants knowing the direction of movement in advance.*Equipment:* Adhesive tape, stopwatch, ball and tennis racquet.*Space:* The test is performed on a tennis court because, for a more accurate measurement, marked lines are needed and a large space is needed for the ball hitting.*Test description:* Participants are waiting for a start signal at the center line on the baseline of the tennis court (Figure 4). With a specific movement for tennis (according to the desire, feeling and skill of movement in a tennis game), they cross the distance to point B (crossing the base line and the side single line) where they hit a vertically dropped ball by the measurer with a forehand, return to the starting point and repeat task after which they return to the middle again. From the middle, the participant moves diagonally to point C of the intersection of the service line and the side single line, where he also has one forehand stroke, moves to the center line on the service line (point D) and strikes the backhand. After the backhand stroke, the sprint crosses diagonally and the racket touches the marked part of the net for the end of the test. Then, the time stops.*Measurer:* Three measurers, one with a stopwatch and the other two dropping the balls on marked places.*Assessment/Evaluation:* The result is the time measured in tenths of a second from the start signal to the end of the task. Three attempts are carried out and all are recorded on a score form.**Tennis-specific reactive agility new (TS-RAN) test** is a test designed to assess specific agility in tennis with a movement pattern that simulates the actual situation in the game, with participants not knowing the direction of movement in advance. They perform two strokes (forehand or backhand) on the baseline and two strokes (forehand or backhand) on the service line.*Equipment:* Adhesive tape, stopwatch, ball and tennis racquet.*Space:* Test is conducted on a tennis court.*Test description*: Participants move from the starting point A in the middle of the base line of the tennis court, in the basic tennis position, to the sound and visual signal from the measure that shows the direction of movement to the point intended for performance (B’ backhand or B forehand). A measurer, who drops the balls vertically, stands at the intended points B’ and B and also follows the visual signs by a third measurer standing on the other side of the net (Figure 5). After the participant has made the first forehand or backhand blow, he returns to the middle of the baseline to the starting position, and following the signs of the measurer on the other side of the net, he/she makes another stroke on the baseline after which he/she returns to the starting position. From point A, the participant then moves diagonally to hit from the service line, depending on the direction shown by the measurer (backhand C’ or forehand C). After the third stroke of the ball on the service line, the participant must come to the middle of the service line and notice the last visual sign for the last forehand or backhand stroke. The last part of the test is carried out by sprinting (from point C’ or point C) to the marked part of the middle of the net (point D). The time is stopped after the participant has touched the marked part of the net with the racket. The measurer standing on the other side of the net shows signs/signals by lifting the outstretched left or right hand sideways, depending on the stroke.*Measurer*: Four measurers, one with a stopwatch, one showing the signals on the opposite side of the net and one measurer on each side of the side single lines that also follow the measurer across the net and throw the balls to the marked places.*Assessment/Evaluation*: The result is the time measured in tenths of a second from the start signal to the end of the task. Three attempts are carried out and all are recorded on a score form.

All tests were conducted on an outdoor tennis court, on hardcourt surface, with participants dressed in sportswear, as they would be in regular training sessions or during competitions. All the participants brought their own tennis rackets to the test, for the tennis-specific agility tests. Testing was conducted in June 2021, with tests being performed for three Tuesdays in a row, at the same time of day (9:00–10:30 a.m.), during a regular sports condition session, which is, for the tested group, regularly scheduled once a week. The outside temperature during the testing ranged from 20–25 °C.

During the first’s week testing session, the two generic agility tests (MDA “ABCD”; STSLA) were performed by the participants. After a week, the same was repeated for the two tennis-specific pre-planned agility tests (TS-STSLA and TS-MDA) and in the last week the new tennis-specific reactive agility test (TS-RAN) was conducted.

For all the tests, one familiarization session/trial was conducted by all the participants, before three official trials were performed with 4–6 min rest periods between the trials and few minutes more between the tests.

### 2.3. Statistical Analysis

All statistical calculations were carried out using data analysis software Statistica 14.0.0.15 (TIBCO Software Inc., 2020, Palo Alto, CA, USA).

Data for all test trials, which included two validated generic agility tests (MDA “ABCD” and STSLA), two tennis-specific pre-planned agility tests (TS-STSLA and TS-MDA) and the new tennis-specific reactive agility test (TS-RAN), were presented as mean ± standard deviation. In order to identify deviations from normal distribution, the Kolmogorov–Smirnov test with Lilliefors correction was applied.

One way within-subject ANOVA was applied to identify systematic bias between trials, and F-value (F) and significance level (*p*) were presented.

Cronbach alpha was calculated as a measure of internal consistency between trials. Type one error was set at α = 5%.

Furthermore, average inter-item correlation (IIR) together with intraclass correlation coefficient (ICC) were calculated as measures of reliability. Average inter-item correlation (IIR) examines the extent to which scores on one item are related to scores on all other items in a scale. Basically, it is correlation between test items. The intraclass correlation coefficient (ICC), which is probably the best suited measure/index of reliability, can reflect both the degree of correlation and agreement between measurements. An ICC of <0.50 was poor, between 0.50 and 0.75 it was moderate, between 0.75 and 0.90 it was good and >0.90 was excellent [22].

To establish the construct/factorial validity of the newly designed tennis-specific agility tests, factor analysis (Guttman–Kaiser criterion of extraction) was conducted for the variables/tests forming the test battery.

The test’s preparation/setting time, the time required to test one participant, the required technical equipment and the number of required measurers conducting the test were the assessment criteria for the test’s overall feasibility.

## 3. Results

The basic descriptive statistical parameters (mean, standard deviation, distribution of the results) for all the trials within the each of the measured tests are presented in Table 1. The results show a normal data distribution pattern for all the test variables, except the multi-direction agility “ABCD” (MDA “ABCD”) test where a minor deviation from the normal distribution of data is noticed.

One way within-subject ANOVA showed systematic bias between trials as significant differences were found between trials throughout the test battery (Table 1). This indicates that there was systematic changes or inconsistency in the performance of individual participants between testing trials.

The reliability of all the agility tests constituting the test battery is also presented in Table 1, using different measures of reliability and internal consistency (Cronbach alpha, IIR, ICC), aiming to address both the within and between-subject reliability.

Internal reliability measures of average inter-item correlation (IIR) and Cronbach alpha (CA) showed a “very good” to “excellent” internal consistency/scale reliability between trials for all the variables (CA 0.89–0.97; IIR 0.75–0.92), with the steps to the side lateral agility test (STSLA) being the only exception, showing values that can be considered “acceptable” internal consistency (CA 0.74; IIR 0.51). The internal consistency between trials for the newly constructed tennis-specific reactive agility new test (TS-RAN) proved to be on a margin between being very good or excellent (CA 0.90; IIR 0.75).

Intraclass correlation coefficient (ICC), which considers both the degree of correlation and agreement between measurements, ranged from 0.48 for the steps to the side lateral agility (STSLA) test, which is considered poor reliability, to the 0.92 for the multi-direction agility “ABCD” (MDA “ABCD”) test, what is excellent test reliability.

However, for all the three tennis-specific agility tests, tennis-specific steps to the side lateral agility (TS-STSLA), tennis-specific multi-direction agility test (TS-MDA) and the tennis-specific reactive agility new test (TS-RAN), the intraclass correlation coefficient (ICC) has shown values that are much more moderately ranged from 0.73 to 0.76, which is a level of reliability that can be considered as “moderate” to “good” (Table 1). The newly constructed tennis-specific reactive agility new test (TS-RAN) showed to be right on the margin of “moderate” to “good” reliability with an ICC of 0.74 (95% CI 0.48–0.92; *p* < 0.01).

The results of the factor analysis (G-K criterion) calculated for all five agility tests showed that one significant latent dimension-factor has been extracted, and that the extracted latent dimension explained 65% of the tests’ common variance (Table 2).

All the tests, except the multi-direction agility “ABCD” (MDA “ABCD”) test have been highly projected on the extracted principal component, with the two newly designed tennis-specific agility tests (TS-MDA; TS-RAN) showing the highest projections of 0.93 and 0.91, respectively.

## 4. Discussion

High-level tennis players must be able to perform rapid, multidirectional movements to be able to adequately place themselves in the best possible position for hitting the ball and making a successful stroke technique [1,23].

Surely, agility is a highly important ability for tennis players and therefore designing adequate instruments to test and monitor its development in different stages of players’ careers is an important task, both for sports scientists and coaches working in the field of tennis.

Before the implementation of any newly developed instrument/test, it should be tested for its measurement properties [24].

Thus, the main aim of this study was to design a tennis-specific reactive agility test, evaluate its measurement properties and compare it to some already validated generic agility tests and some other newly constructed tennis-specific tests.

### 4.1. Reliability of Measurements

One of the first steps in the procedure of checking test applicability is to check its reliability, and both within-subject reliability and between-subject reliability were tested, as both are important indicators of the overall test quality.

The results obtained by using different reliability indexes revealed that the between-subject reliability of the used agility tests was more than satisfactory, as indicated by the values of Cronbach alpha and IIR (Table 2), ranging from very good to excellent (CA 0.89–0.97) for the whole test battery, except the generic steps to the side lateral agility (STSLA) test. The STSLA test is shortest in duration and in case of this sample of participants it showed to be the least sensible, as within the relatively small result margins many of the participants could not retain their place at the measuring scale between the three trials.

The newly constructed tennis-specific reactive agility new test (TS-RAN) showed to be internally very reliable/consistent as its between-subject reliability proved to be quite high (CA 0.90) and its reliability values almost identical to the other two tennis-specific agility tests (TS-STSLA and TS-MDA).

However, the within-subject reliability of the used test showed a systematic bias between trials, as indicated by the results of within-subject ANOVA presented within Table 1. This bias points to inconsistency in performance of each participant between testing trials and can probably be explained with the learning effect, which is clearly visible in the systematic improvement of the results between the individual trials (Table 1). It can also partially be caused by the very young age of the participants (10–12 years), as they showed a lack of performance consistency in each trial execution.

As a result, the intraclass correlation coefficient (ICC), which besides the degree of correlation between trials also considers agreement between measurements in its calculation, showed values that were not so high as was the case with the between-subject reliability. The newly constructed tennis-specific reactive agility new test (TS-RAN) showed almost similar values of the ICC to the other two tennis-specific agility tests (TS-STSLA, TS-MDA), achieving an acceptable level of reliability that can be considered “moderate” to “good” (Table 2). These results are also completely in line with the results of the other recently presented tennis-specific reactive agility test (TAT) [17]. The TAT test has also shown moderate relative reliability with an ICC value of 0.74, what is exactly the same as for the TS-RAN test.

Overall, concerning the reliability of the newly constructed tennis-specific reactive agility new test (TS-RAN), it can be pointed out that although some systematic changes between testing trials were found, those changes did not significantly impact the between-subject reliability. Participants mostly retained their relative position on the measuring scale in comparison with the other participants of the research, but mostly affected the within-subject reliability, as individual performances of each subject significantly varied between trials.

However, although the reliability of the newly constructed tennis-specific reactive agility new test (TS-RAN) was affected by the observed measurement error, its overall reliability can still be considered satisfactory (“moderate” to “good”) and is basically identical to the values of the other two tennis-specific agility tests (TS-STSLA, TS-MDA).

The noticed inconsistency in performance of each participant between testing trials is not quite unexpected, as in circumstances when significant differences between testing trials are found, it is often the within-subject reliability that is questionable [25].

### 4.2. Factorial Validity

Factor analysis was applied on the all five agility tests, two generic and three tennis-specific ones. One of the tests (TS-RAN) includes reactive agility in form of cognitive reaction to a visual stimulus and the other two (TS-STSLA, TS-MDA) were pre-planned change of direction (COD) tests but performed in a real tennis game environment, simulating conditions during a tennis match.

The results of the applied factor analysis showed that a single latent dimension (factor), which explains 65% of the common variance, has been identified and that all tests except the multi-direction agility “ABCD” (MDA “ABCD”) test have been highly projected on the extracted principal component. Therefore, all the agility tests used in this study could claim to be valid measures of agility and could claim to measure the same construct to a greater or lesser extent.

The three tennis-specific agility tests (TS-STSLA; TS-MDA; TS-RAN), which were performed on a tennis court with tennis equipment, simulating the real tennis game situations, showed the highest projections on the common latent dimension of 0.84, 0.93 and 0.91, respectively. It is obvious that the more complex tennis-specific agility movement patterns, designed to imitate real tennis game conditions, explain the common dimension variance to a greater extent than the generic agility tests.

Of the two performed generic agility tests, both commonly used in general sport populations and are validated, the steps to the side lateral agility test (STSLA) was the one that showed a relatively high projection on the factor (0.79), while the multi-direction agility “ABCD” (MDA “ABCD”) test showed a relatively low projection (0.43). That is not surprising as the STSLA test, although a validated generic test, has a movement pattern that resembles the lateral movement of tennis players while playing from the baseline. Therefore, although generic in nature and performed without tennis equipment, the STSLA test could partially be considered a tennis-specific test as well, which can probably explain its relatively high projection on the common factor dominated by tennis-specific agility tests.

Relations between tennis-specific and generic agility tests were partially addressed in the paper published by Jensen at al. (2021) [17], where concurrent validity results showed a significant positive moderate correlation between the tennis-specific reactive agility TAT test and the already validated generic Spider Drill [14]. However, further research is needed to gain a deeper understanding as to whether increasing the sport specificity of the agility tests will make a significant difference in better predicting a future tennis performance.

It is also worth mentioning that, based on the individual correlations of the tennis-specific agility tests with the common latent dimension/factor, it was not possible to discriminate between the two tennis-specific agility tests, which use the pre-planned movement pattern (TS-STSLA; TS-MDA), and the tennis-specific reactive agility new test (TS-RAN), which has been designed to measure agility in a “non-planned” conditions. In TS-RAN, test participants have to react to a visual signal and pick the signaled movement heading from the two possible test directions, which are equally long. Therefore, the TS-RAN is supposed to measure both the cognitive and physical aspects of agility, while the other two tennis-specific tests (TS-STSLA; TS-MDA) are basically COD tests, performed in tennis-specific conditions and designed to cover the physical side of agility.

Therefore, some differences in measuring properties between the pre-planned- and non-planned-specific agility tests could have been presumed to exist. However, that did not prove to be the correct assumption/hypothesis in the case of this research study and for this sample. On the other hand, because of their involvement in tennis, the participants have already developed sport-specific agility to a certain extent. Therefore, a superior agility performance in one of the tennis-specific tests could easily be “transferred” to a superior agility performance in other specific agility tests, even if they are just tennis-specific COD tests or they are designed to measure reactive “non-planned” agility.

Whatever the reason, the tennis-specific reactive agility new test (TS-RAN) proved to have a good factorial validity in relation to this sample of tests, as it was highly projected (0.91) on the latent dimension which is predominately explained by the tennis-specific agility tests.

### 4.3. Feasibility of the Newly Constructed Tennis-Specific Reactive Agility Test

In terms of the required equipment, the tennis-specific reactive agility new (TS-RAN) test only requires standard tennis equipment (tennis racket and balls) and a stopwatch. In addition, there is a great convenience of using the already existing lines of the tennis court and their intersection points. Thus, no time is needed for test preparation and no additional equipment (measuring meter, adhesive tape, cones etc.) is required for its set-up. Regarding the place and space required for the test implementation, its set-up and the equipment needed, the test is extremely feasible and convenient to be conducted during regular training sessions without any special demands to be put on the test organizers.

The only major demand to be put on the test organizer is that there are four assistants needed for the test to be properly conducted:One to give visual signals to players, pointing in the direction they should move towards;Two to drop tennis balls at the appropriate tennis line intersection points;One to measure the time with a stopwatch and write the results on the pre-prepared form.

However, if the sample of participants consists of players who are in junior ranks or older, it should not be a problem assigning test assistants to some of the participants. However, along with giving them detailed instructions, it is recommended to also give them a number of test trials, for them to feel the right rhythm and timing of dropping the tennis balls and giving visual signals at appropriate moments.

Considering the practical implications of the newly designed tennis-specific reactive agility test and discussing its possible limitations with experts, not many disadvantages have been emphasized. The main potential “issue” which can influence the test results’ reliability, is a human factor, which is usually the case. The test conductors have to coordinate between themselves and “catch”/“feel” the right timing for dropping the tennis balls at the required spots at an appropriate moment, as it is important that the whole test procedure runs fluently, with participants “running into a shot” instead of waiting or being late for it. It is also important that the test conductor who is standing at the opposite side of the court and gives visual signals to players pointing in the direction they should move towards, to do so by constantly using different movement patterns between the trials, preventing participants from getting used to one pattern. However, all those issues are easily solvable, as based on the experience gained during the testing, the test conductors were able to perform testing in a manner required after only a few test trials.

Not using sophisticated measuring equipment (e.g., photo cells) can certainly be considered a disadvantage, as the manual measurement of the test time with a stopwatch is certainly less reliable than using photo cells for instance. However, the fact that the test does not require any sophisticated technical equipment and can use the existing features of the tennis court is at the same time the test’s main advantage, which makes it extremely feasible and practical for being regularly implemented, either during regular training sessions or organized testing trials.

## 5. Conclusions

In conclusion, it is possible to say that the newly constructed tennis-specific reactive agility new (TS-RAN) test generally shows promising results, which are very much in line with the results of the other recently presented tennis-specific reactive agility tests (TAT) [17].

The measurement characteristics of the newly designed reactive agility test are satisfactory and are in line with the other two tennis-specific agility tests that were used in this study.

The new test’s overall reliability can only be considered “moderate to good”, as although the results of participants showed really good internal consistency, the within-subject reliability of the test proved to be unsatisfactory, as the participants showed a lack of performance consistency. However, as that was the case with the other tests within the used test battery, that can be attributed to the sample of the participants. The lack of performance consistency is not unexpected [24], especially considering the very young age of participants who performed the test for the very first time.

The newly designed reactive agility test had good factorial validity, as it was highly projected on the same single latent dimension, with variance predominately explained by the tennis-specific agility tests. However, although the TS-RAN test was designed to measure both the cognitive and physical aspects of agility, in this sample of participants, it did not show its ability to differentiate itself from the other tennis-specific agility tests (TS-STSLA; TS-MDA) that were designed to only cover the physical side of agility.

The test’s greatest strength is its feasibility, which is high since the test does not require any specific set-up nor technical equipment, which makes it practical for being regularly implemented in a practical setting.

This research was conducted as a small-scale pilot study, aiming to evaluate the test’s design and potential for future use, and it has provided a first insight into the test’s measurement properties and its potential. The newly constructed tennis-specific reactive agility new test (TS-RAN) has similar qualities to the other recently presented tennis-specific reactive agility tests (TAT) [17]. However, it is even more feasible and practical for regular use during training and testing sessions; nevertheless, a large-scale study on a larger sample of examinees that consists of older and more advanced tennis players would certainly be required to determine whether increasing the sport specificity of the agility tests makes a significant difference in better predicting future tennis performance. Only then would be possible to fully assess the test’s potential to be widely accepted and used.

## Figures and Tables

**Figure 1 ijerph-19-10039-f001:**
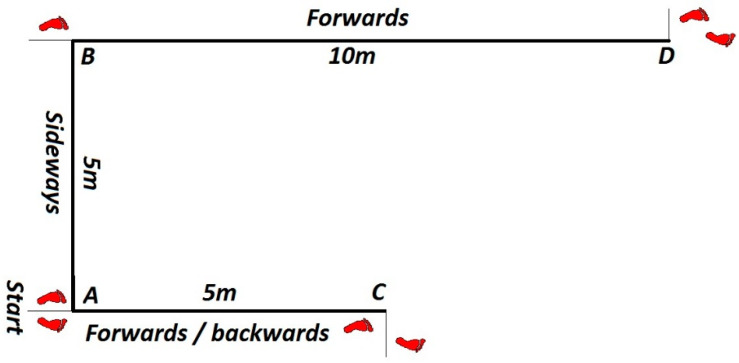
Multi-direction agility test ACABD set-up.

**Figure 2 ijerph-19-10039-f002:**
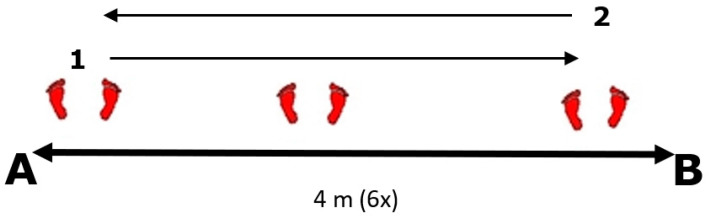
Steps to the side lateral agility test [20].

**Figure 3 ijerph-19-10039-f003:**
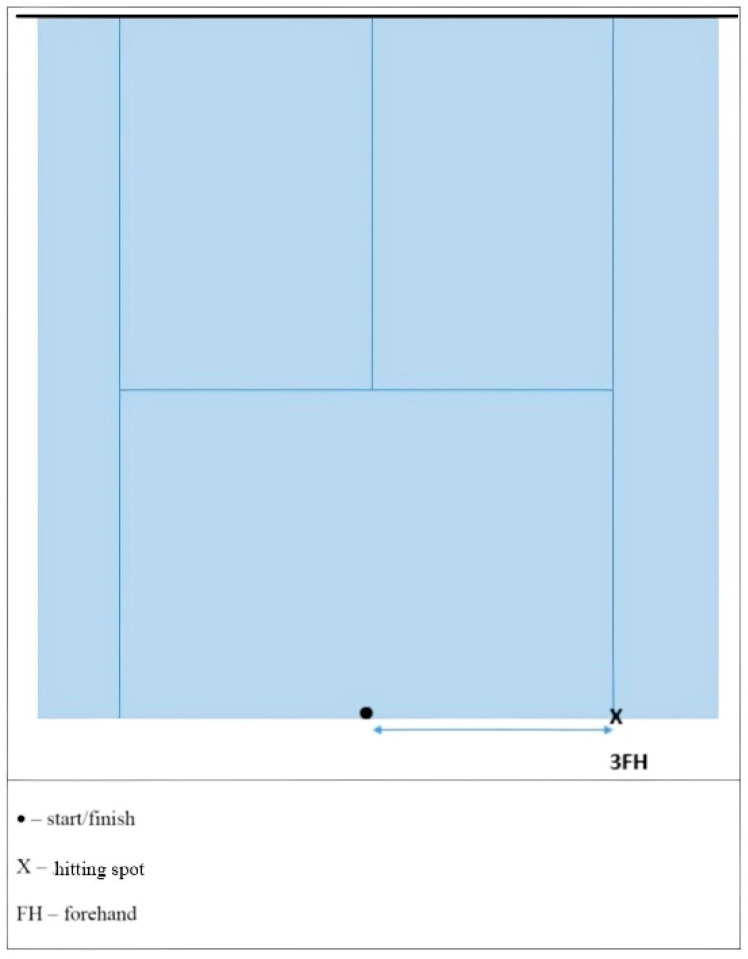
Tennis-specific steps to the side lateral agility (TS-STSLA) scheme.

**Figure 4 ijerph-19-10039-f004:**
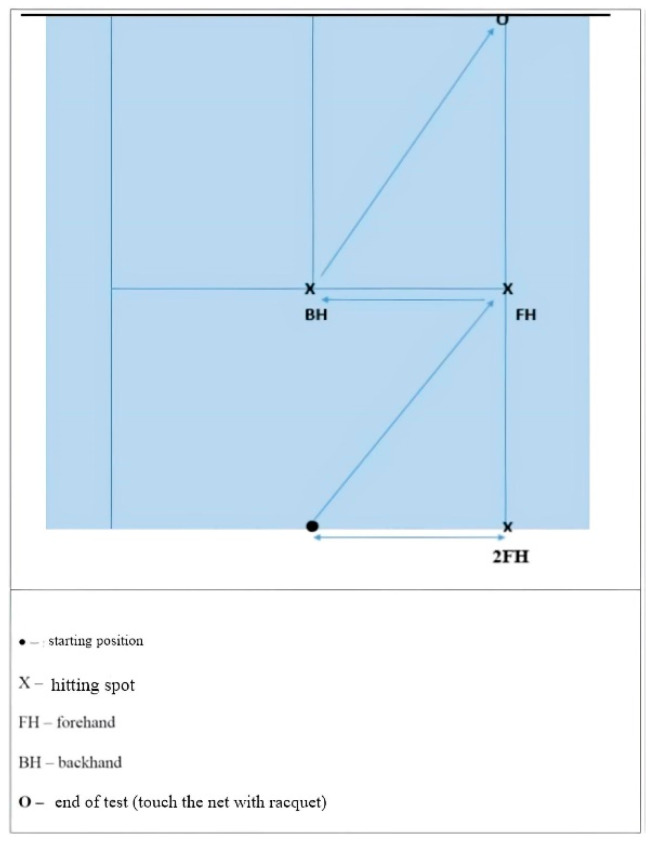
Tennis-specific multi-directional agility (TS-MDA) test scheme.

**Figure 5 ijerph-19-10039-f005:**
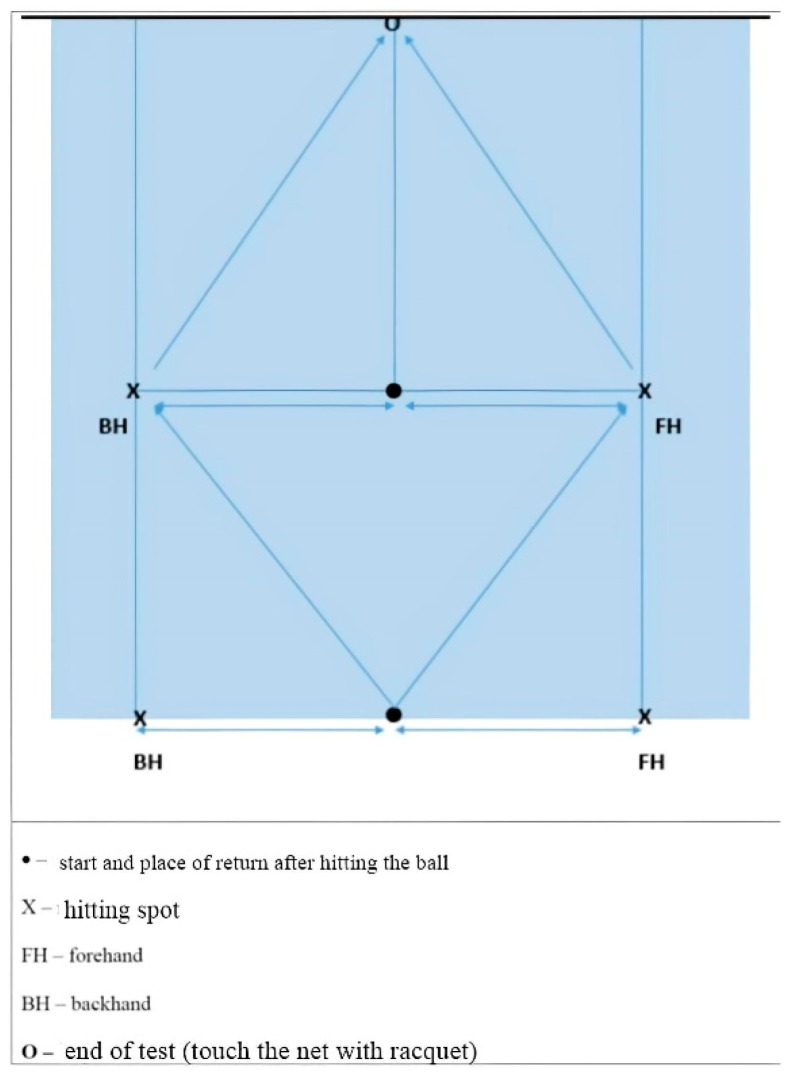
Tennis-specific reactive agility new (TS-RAN) test scheme.

**Table 1 ijerph-19-10039-t001:** Basic descriptive parameters and reliability analyses of different agility tests used within the test battery.

	Trial 1	Trial 2	Trial 3	F	*p*	ICC	IIR	Cα	KS Test
MDA “ABCD”	10.78 ± 1.25	10.50 ± 1.27	10.35 ± 1.22	11.63	**<0.01**	0.92	0.92	0.97	***p* < 0.05**
STSLA	10.08 ± 0.81	9.87 ± 0.54	9.77 ± 0.61	3.17	**0.05**	0.48	0.51	0.74	*p* > 0.20
TS-STSLA	10.10 ± 0.92	9.57 ± 0.63	9.26 ± 0.79	35.87	**<0.01**	0.76	0.79	0.90	*p* > 0.20
TS-MDA	12.15 ± 0.82	11.86 ± 0.79	11.80 ± 0.94	5.30	**<0.01**	0.73	0.75	0.89	*p* > 0.20
TS-RAN	14.71 ± 1.35	14.23 ± 1.15	14.25 ± 1.56	4.68	**0.01**	0.74	0.76	0.90	*p* > 0.20

MDA “ABCD”—Multi-direction agility “ABCD” test; STSLA—Steps to the side—lateral agility; TS-STSLA—Tennis-specific steps to the side lateral agility; TS-MDA—Tennis-specific multi-direction agility; TS-RAN—Tennis-specific reactive agility new; F—Value on the F distribution, used to determine whether the differences between the test trials are statistically significant; ICC—Inter-class correlation coefficient; IIR—Inter-item correlation; Cα—Cronbach alpha; **Bold type** has been used to mark statistically significant differences.

**Table 2 ijerph-19-10039-t002:** The results of the factor analysis (G-K criterion) calculated for all five agility tests.

Variable/Test	Extraction: Principal Components
	Factor (F1)
MDA “ABCD”	−0.43
STSLA	**−0.79**
TS-STSLA	**−0.84**
TS- MDA	**−0.93**
TS-RAN	**−0.91**
**Expl.Var**	3.23
* **Prp.Totl** *	0.65

MDA “ABCD”—Multi-direction agility “ABCD” test; STSLA—Steps to the side lateral agility; TS-STSLA—Tennis-specific steps to the side lateral agility; TS-MDA—Tennis-specific multi-direction agility; TS-RAN—Tennis-specific reactive agility new; Expl. Var.—Explained variance; Prp.Totl—Proportion of total variance explained; Factor (F)—Correlations of the tests with the main component of factor analysis. **Bold type** has been used to mark statistically significant differences.

## Data Availability

Not applicable.

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
