# Peer review of "Design and Validation of a New Tennis-Specific Reactive Agility Test—A Pilot Study"

_ijerph, 2022, doi:10.3390/ijerph191610039_

Round 1

Reviewer 1 Report

The study is about the design and validation of a new tennis-specific reactive agility test that would be feasible and practical for regular implementation. The results showed that the overall reliability of the test was "moderate to good", as although the results of participants showed really good internal consistency.

Manuscript is correctly written and clearly justifies the importance of novelty of the study from the great importance of agility in tennis, a sport in which rapid changes of direction and high demands for technical-tactical adaptation, together with the requirements of the coordinative abilities are continuous. It is very important to design specific tests that help to know the agility levels of tennis players for better training planning.

However, there are some important issues to correct or to improve the document that I indicate below

- In “Keywords” I advise authors to delete "reactive agility" and "tennis-specific", which are already in the title, and include other synonymous words to make it easier for other researchers interested in your line of research to find your article.

- Authors should make a significant effort to strengthen the introduction. In addition to the bibliographic references of the document, some more current references should be included. The most current reference is from 2017. There are references closely related to the study that are also very current. Thus, I indicate four very current scientific publications in case they can help the authors to improve the introduction and discussion of the results:

Jansen, M. G. T.; Elferink-Gemser, M. T.; Hoekstra, A. E.; Faber, I. R.; Huijgen, B. C. H., Design of a Tennis-Specific Agility Test (TAT) for Monitoring Tennis Players. J. Hum. Kinet. 2021, 80, (1), 239-250.

Baiget, E.; Iglesias, X.; Fuentes, J. P.; Rodriguez, F. A., New Approaches for On-court Endurance Testing and Conditioning in Competitive Tennis Players. Strength and Conditioning Journal 2019, 41, (5), 9-16.

Kilit, B.; Arslan, E.; Soylu, Y., Effects of different stretching methods on speed and agility performance in young tennis players. Science & Sports 2019, 34, (5), 313-320.

In "Materials and Methods", section "Participants" the authors describe that the sample consisted of 32 youth tennis players (21 males and 11 females; 10.85 ± 1.50 years), being important that they justify how the possible differences between males and females and a significant difference between the number of males (21) versus females (11) could influence the results. In case you cannot justify in the method that this question does not influence the results, you should in the "Discussion" section, talk about this limitation,

In "Materials and Methods" I recommend improving "Figure 2. Steps to side – lateral agility test", I think it belongs to other authors (Metikoš et al., 1989) and, furthermore, the distances are written by hand, and the person on the left looks like a person different. Actually figure 2 is of very low quality and is not specific to the new test.

- In the "Discussion", I suggest that the authors consider the same thing that I indicated in the "Introduction" about new citations. Authors should include some more citations that give this section a higher quality.

- The "Conclusions" section must be refined taking into account what has been commented in relation to the results and the discussion.

Author Response

Comments and Suggestions for Authors

The study is about the design and validation of a new tennis-specific reactive agility test that would be feasible and practical for regular implementation. The results showed that the overall reliability of the test was "moderate to good", as although the results of participants showed really good internal consistency.

Manuscript is correctly written and clearly justifies the importance of novelty of the study from the great importance of agility in tennis, a sport in which rapid changes of direction and high demands for technical-tactical adaptation, together with the requirements of the coordinative abilities are continuous. It is very important to design specific tests that help to know the agility levels of tennis players for better training planning.

However, there are some important issues to correct or to improve the document that I indicate below

- In “Keywords” I advise authors to delete "reactive agility" and "tennis-specific", which are already in the title, and include other synonymous words to make it easier for other researchers interested in your line of research to find your article.

 The authors made the rightly suggested changes and have replaced the keywords with the new ones. Thanks.

- Authors should make a significant effort to strengthen the introduction. In addition to the bibliographic references of the document, some more current references should be included. The most current reference is from 2017. There are references closely related to the study that are also very current. Thus, I indicate four very current scientific publications in case they can help the authors to improve the introduction and discussion of the results:

Jansen, M. G. T.; Elferink-Gemser, M. T.; Hoekstra, A. E.; Faber, I. R.; Huijgen, B. C. H., Design of a Tennis-Specific Agility Test (TAT) for Monitoring Tennis Players. J. Hum. Kinet. 2021, 80, (1), 239-250.

Baiget, E.; Iglesias, X.; Fuentes, J. P.; Rodriguez, F. A., New Approaches for On-court Endurance Testing and Conditioning in Competitive Tennis Players. Strength and Conditioning Journal 2019, 41, (5), 9-16.

Kilit, B.; Arslan, E.; Soylu, Y., Effects of different stretching methods on speed and agility performance in young tennis players. Science & Sports 2019, 34, (5), 313-320.

 The authors included some of the suggested references, of which especially the first one (Design of a Tennis-Specific Agility Test (TAT) for Monitoring Tennis Players. J. Hum. Kinet. 2021, 80, (1), 239-250.) is extremely relevant to our paper findings. Therefore, the same was added not only to the Introduction, but to the Results and Discussion sections of the manuscript as well.

It is funny that some of the authors of that paper are my dear friends and colleagues from some of international sport science committees and boards and I manage to skip on their work. Shame on me ☹. The reason for it is that our paper had been written during the last summer, but the authors couldn’t find time to prepare it for publishing before just recently… and in the meantime the above-mentioned paper written by the Dutch group of authors was published.

So, the lesson taken is, that if a paper has not been published soon enough after writing it, the authors should update the references before publishing it ?.

In "Materials and Methods", section "Participants" the authors describe that the sample consisted of 32 youth tennis players (21 males and 11 females; 10.85 ± 1.50 years), being important that they justify how the possible differences between males and females and a significant difference between the number of males (21) versus females (11) could influence the results. In case you cannot justify in the method that this question does not influence the results, you should in the "Discussion" section, talk about this limitation,

 The above-mentioned gender issue was addressed and added to the Participants section of the manuscript.

In "Materials and Methods" I recommend improving "Figure 2. Steps to side – lateral agility test", I think it belongs to other authors (Metikoš et al., 1989) and, furthermore, the distances are written by hand, and the person on the left looks like a person different. Actually figure 2 is of very low quality and is not specific to the new test.

 The figure 2. was removed and replaced with a new one, as suggested.

- In the "Discussion", I suggest that the authors consider the same thing that I indicated in the "Introduction" about new citations. Authors should include some more citations that give this section a higher quality.

Some more citations/references were added to the Discussion section of the manuscript, as advised. The authors focused primarily on papers, which are very similar in scope and methods to our paper, and which results have a direct impact on it.

- The "Conclusions" section must be refined taking into account what has been commented in relation to the results and the discussion.

The Conclusions section has been refined to be more in line with the findings that were presented in the Results and Discussion sections of the manuscript.

Submission Date

16 June 2022

Date of this review

25 Jun 2022 18:39:14

Reviewer 2 Report

The authors developed a new agility test for tennis players that considered the inclusion of both physical and cognitive agility of tennis players. The authors compared the measurement properties of the newly developed test with the other existing tests. The test was reported to be somewhat reliable with a good factorial validity of 0.91 explaining a single dimensionality of the test. Overall, the study is timely considering the lack of specific measures of cognitive and agility functions of tennis players, and I congratulate the authors for their effort in developing the test. However, many issues hinder the publication of the manuscript in its current form.

1.       L12 in the abstract (10,85 ± 12 1.50 years), is it supposed to be (10.85 ± 12 1.50 years)? A point, not a comma. The same issue appeared under the participant’s section.

2.       Why were 10-12 years old considered as participants for this study?

3.       Do normality tests carry out to ensure that the players are of the same levels of performance prior to all tests?

4.       L232-234 the variables and test batteries should be indicated

5.       Was there any familiarization session conducted before the commencement of the full data collection?

6.       I think the tests should have been carried out on a different day in a relatively similar condition. The authors implemented the tests by giving three trials to the participants with only 4-6m rests. The trials coupled with all the other tests were conducted on the same day (L206-213). I think this approach may bring about fatigue (depending on their fitness level) which could affect their actual performance. Moreover, conducting the test within a single time frame may not permit the projection of its test-retest reliability at a varying time which is essential for any newly developed test.

7.       The validity measurement of the test is highly debatable. I am wondering why the authors compared and tested the validity of the TS-RAN with the other tests when it is demonstrated that the tests are not similar in both protocols and objectives. To measure the factorial validity of the test, I think the authors ought to extract the psychometric properties of the test such as explosive power, speed movements, dynamic balance, coordination etc and then introduce a factor analysis to determine the effectiveness of the test in measuring the aforementioned elements.

8.       The discussion section is mainly developed on speculations. No adequate literature/references were cited to support the findings of the study.

Author Response

Comments and Suggestions for Authors

The authors developed a new agility test for tennis players that considered the inclusion of both physical and cognitive agility of tennis players. The authors compared the measurement properties of the newly developed test with the other existing tests. The test was reported to be somewhat reliable with a good factorial validity of 0.91 explaining a single dimensionality of the test. Overall, the study is timely considering the lack of specific measures of cognitive and agility functions of tennis players, and I congratulate the authors for their effort in developing the test. However, many issues hinder the publication of the manuscript in its current form.

  1. L12 in the abstract (10,85 ± 12 1.50 years), is it supposed to be (10.85 ± 12 1.50 years)? A point, not a comma. The same issue appeared under the participant’s section.

The technical issue has been fixed. Thanks.

  1. Why were 10-12 years old considered as participants for this study?

The data were collected by one of my students, for her master thesis, which I had mentored. She works as a coach in the TC “Split”, the best-known tennis club in the country, as four former TOP 10 players came out of its tennis school, including a former Wimbledon winner, Goran Ivanišević.

The10-12 years old were considered as participants, as only in the youth categories it was possible to find a big enough sample to conduct the research. Moreover, the selected sample included participants who were in young prepuberty/early puberty age, where gender related inputs still haven’t reached their full swing (as shown by the test results, which have not shown any statistically significant differences between genders). So, as the gender related issue did not show to have any significant influence on the test results for this particular age group, that allowed the researchers to add some girl players of the same age to the predominantly boys sample and to slightly enlarge it.

  1. Do normality tests carry out to ensure that the players are of the same levels of performance prior to all tests?

The normality test was carried out for all the variables, with only (MDA “ABCD”) test showing minor deviation from the normal distribution.

  1. L232-234 the variables and test batteries should be indicated

The authors have made the suggested inputs

  1. Was there any familiarization session conducted before the commencement of the full data collection?

For all the tests, one familiarization session/trial was conducted by all the participants, before three official trials were performed. The familiarization trial has been important not only for the players, but for the measurers as well to synchronize between themselves and to catch the right timing for dropping the tennis balls. Therefore, the authors believe it should become a regular part of the testing procedure for the new test.

  1. I think the tests should have been carried out on a different day in a relatively similar condition. The authors implemented the tests by giving three trials to the participants with only 4-6m rests. The trials coupled with all the other tests were conducted on the same day (L206-213). I think this approach may bring about fatigue (depending on their fitness level) which could affect their actual performance. Moreover, conducting the test within a single time frame may not permit the projection of its test-retest reliability at a varying time which is essential for any newly developed test.

The tests were carried out in three different days (for three Tuesdays in a row) and in relatively similar conditions (month of June; similar temperature, humidity etc), at the same time of day (9:00-10:30 AM). That information has now been added to the manuscript, in its Measures/Variables section (L227-235).

  1. The validity measurement of the test is highly debatable. I am wondering why the authors compared and tested the validity of the TS-RAN with the other tests when it is demonstrated that the tests are not similar in both protocols and objectives. To measure the factorial validity of the test, I think the authors ought to extract the psychometric properties of the test such as explosive power, speed movements, dynamic balance, coordination etc and then introduce a factor analysis to determine the effectiveness of the test in measuring the aforementioned elements.

The authors disagree with the remark above. The one of the study’s aims hasn’t been to determine the extent of influence/correlation of the different factors (such as explosive power, speed movements, dynamic balance, coordination etc) with the new test, but to establish correlation of the new test with some already established and validated generic agility tests and some tennis specific ones. So, the aim was to establish how our newly designed tennis-specific agility test “lines-up” to some already validated and used agility tests and to which extent they measure the same latent construct.

  1. The discussion section is mainly developed on speculations. No adequate literature/references were cited to support the findings of the study.

The Discussion section has been slightly refined, by adding some new references that should support our study findings. The authors focused primarily on papers, which are same/similar in scope and methods to our paper, and which results have had a direct impact on our paper’s findings.

Round 2

Reviewer 1 Report

The authors have answered all the questions correctly and have substantially improved the manuscript.

Author Response

The authors very much appreciate the reviewer's efforts to provide us with valuable suggestions that have surely enhanced the overall quality of the manuscript. 

Reviewer 2 Report

The manuscript has improved from the previous iteration.

However, my main concern still remains unanswered.  I am wondering why the authors compared and tested the validity of the TS-RAN with the other tests when it is demonstrated that the tests are not similar in both protocols and objectives.

The author stated "So, the aim was to establish how our newly designed tennis-specific agility test “lines-up” to some already validated and used agility tests and to which extent they measure the same latent construct". The question is how could the specific test developed in the current study possibly have a similar latent construct with the other tests when the other tests are not similar both in objective and protocols?

Author Response

The authors very much appreciate the both reviewers’ efforts to provide us with suggestions that have surely enhanced the overall quality of the manuscript. The authors have tried to address all the remarks/suggestions wherever it was possible, and as almost all of the suggestions were justified and constructive, it really wasn’t a problem to do so.

However, regarding the last remaining concern of the reviewer 2., the authors do not quite understand why the reviewer considers comparing the newly developed test with some other already validated tests, all hypothesised to measure the same dimension, as a problem?

Testing hypothesis that the new measure/test correlates with some other measures/tests of a similar characteristic, is a completely legitimate research approach… and exploratory factor analysis has traditionally been used to explore the possible underlying structure of a set of interrelated variables without imposing any preconceived structure on the outcome (Child, 1990).

So, by performing exploratory factor analysis the authors aimed to identify the number of latent constructs and the underlying factor structure of a set of variables/tests hypothesised to measure agility… and the results has confirmed that relationship between the observed variables/tests and their underlying latent construct(s) exists.

The factor structure was confirmed, as all the tests are related to the same factor/construct and the hypothesis is obviously supported by the results, at least for the sample of players participating in this pilot study.

Of course, the fact that the tests are relatively highly projected on the same factor and are related to the same construct, still doesn’t automatically mean that the construct is “agility performance”, but there is a rational behind assuming so.

All in all, the authors cannot quite understand why the specific test developed in the current study could not possibly have a similar latent construct/structure with the other used tests, which are similar in both objectives and protocols?... and even if they aren’t similar in objectives and protocols, why would that mean that the results can’t support the hypothesis that they actually measure the same thing/underlying latent construct, whatever we might call it.